# Waveguide coupled III-V photodiodes monolithically integrated on Si

Pengyan Wen [1], Preksha Tiwari [1], Svenja Mauthe[1], Heinz Schmid [1], Marilyne Sousa[1], Markus Scherrer[1], Michael Baumann[2], Bertold Ian Bitachon [2], Juerg Leuthold[2], Bernd Gotsmann [1] & Kirsten E. Moselund[1] ✉

The seamless integration of III-V nanostructures on silicon is a long-standing goal and an important step towards integrated optical links. In the present work, we demonstrate scaled and waveguide coupled III-V photodiodes monolithically integrated on Si, implemented as InP/In$_{0.5}$Ga$_{0.5}$As/InP p-i-n heterostructures. The waveguide coupled devices show a dark current down to 0.048 A/cm$^2$ at −1 V and a responsivity up to 0.2 A/W at −2 V. Using grating couplers centered around 1320 nm, we demonstrate high-speed detection with a cutoff frequency f$_{3dB}$ exceeding 70 GHz and data reception at 50 GBd with OOK and 4PAM. When operated in forward bias as a light emitting diode, the devices emit light centered at 1550 nm. Furthermore, we also investigate the self-heating of the devices using scanning thermal microscopy and find a temperature increase of only ~15 K during the device operation as emitter, in accordance with thermal simulation results.

---

[1] IBM Research Europe – Zurich, Säumerstrasse 4, 8803 Rüschlikon, Switzerland. [2] ETH Zürich, Institute of Electromagnetic Fields (IEF), Gloriastrasse 35, 8092 Zürich, Switzerland. ✉email: kirsten.moselund@epfl.ch

As the amount of data generated from modern communication applications such as cloud computing, analytics, and storage systems is increasing rapidly, silicon electronic integrated circuits (ICs) are suffering from a bottleneck at the interconnection level resulting from the resistive interconnect[1,2]. Electrons are ideal for computation because they allow for ultimately scaled logic gates that can be integrated in a massively parallel fashion using modern CMOS technologies. Photons on the other hand are ideal for transmission because this can be done almost loss-less on chip-size length scales. To get the best of both worlds, it has therefore been a long-standing goal to combine electronics and photonics on a silicon chip, and the distances over which optical on-chip signal transmission may become favorable are also slowly decreasing[3], bringing on-chip optical communication schemes closer.

At the length scales of optical chips, cross-talk may be avoided and the coherence of optical signals may be exploited to allow for transmitting signals of different wavelengths without interference as in wavelength-division-multiplexing (WDM) to increase bandwidth. State-of-the-art high-speed germanium (Ge) photodetectors showing high bandwidth of 100 GHz have been demonstrated[4]. However, relatively high dark currents of Ge detectors may lead to a low signal-to-dark current ratio[5] and more importantly, the indirect band gap of Ge prevents efficient light emission. Thus, for an on-chip integrated photonic link there is a need to integrate alternative materials such as III–Vs to provide the active gain needed for emission. Beyond classical on-chip optical interconnect, there is also a rising interest in highly scaled integrated photonic components, notably single-photon detectors and emitters for applications in quantum computing[6,7].

High-performance on-chip detectors and lasers have been demonstrated based on bonding of a III-V laser stack including quantum wells on top of a silicon wafer with pre-fabricated waveguides and passives[8–10]. The beauty of wafer-bonding is that the full III–V stack is grown lattice matched on an InP substrate and then transferred onto the silicon photonics wafer either as individual chiplets or as a full wafer[11,12], this allows for perfect material quality. However, the active III–V material is generally integrated on top of the photonics integrated circuit (PIC) and therefore necessitates to couple the light evanescently back and forth from the III–V active material on top to the silicon underneath. Whereas photonic devices will inevitably be larger than electronic ones, today's state-of-the-art high-performance integrated photonic components tend to be orders of magnitude larger measuring 100 s of micrometers. To reduce the RC time constant, which is an important factor for power consumption, devices need to be scaled down. In the past few years, research efforts have therefore been focused on the development of scaled hybrid III-V/Si nanophotonic devices, including nanowire photodetectors[13,14], nanowire light sources[15–17], and photoconductors acting as both detectors and emitters coupled by a polymer waveguide[18]. Although high crystal quality III–V nanowires can be achieved and used for devices, the vertical geometry from a device fabrication perspective requires additional engineering and pick-and-place methods for on-chip integration and waveguide coupled solutions[19,20]. These limitations may be overcome with template-assisted selective epitaxy (TASE)[21–23].

When growing III–V directly on Si, defects will arise at the hetero-interface due to the lattice mismatch. Traditionally, they can be gradually filtered out by buffer layers and defect-stopping-layers as it is common in direct InP-based epitaxy on Si[24], or they can be mediated by growth from trenches exposing (111) facets[25]. Using these methods excellent devices have been demonstrated, but integration with waveguides remains difficult.

In nanowire growth one relies on a small interface for nucleation between the III–V and Si[26,27], where defects remain confined near the interface and no propagating dislocations are formed, whereas stacking faults and twins are quite common in nanowire growth. TASE growth is similar to nanowire growth in that we limit the nucleation site to avoid dislocations, whereas the geometry is determined by the template design rather than the growth conditions. The defects in our TASE grown structures can be localized to the small interface between the Si and III–V seed and result in high-quality III-V elsewhere in the template. This is also confirmed by extensive transmission electron microscopy (TEM) investigations, where we generally do not observe dislocations in the studied p-i-n structures. The grown materials are mono-crystalline with an epitaxial relationship to the Si seed. Stacking faults are common, but these should have a minor impact on electrical and optical properties.

The high quality of the material has been demonstrated originally for electronics applications where we could measure mobilities comparable with other III–V films[28,29], and more recently for monolithic optically pumped InP emitters with performance comparable to that of identical devices fabricated by direct wafer bonding[30]. IBM colleagues previously demonstrated the successful transfer of the TASE concept to an advanced 200 mm process line within IBM, where it was used to demonstrate nanosheet InGaAs FinFETs on Si with a 10-nm channel thickness and state-of-the-art performance[31]. The successful integration of logic devices based on the same technology and for large wafer scale, is promising for also achieving large-scale integration for photonic integrated circuits in the future.

In the present work, we demonstrate waveguide coupled devices with grating couplers centered at 1320 nm. We study two different device architectures based on either a straight or a T-shape architecture. In addition, we implemented a double heterostructures (n-InP/i-InGaAs/p-InP/p-InGaAs) to improve carrier confinement. The improved electrostatics enables the investigation of the emission properties when operated as a light-emitting diode (LED). Thermal characteristics of the devices are important factors for performance and reliability evaluation. We investigate the in-situ temperature profile by scanning thermal microscopy (SThM) and establish that the associated temperature increase is within the acceptable range for device operation[32]. We demonstrate, to the best of our knowledge, the first monolithic heterostructure photodetector directly coupled in-plane to a Si waveguide and demonstrating high-speed performance with a 3 dB frequency of 70 GHz. Using grating couplers centered around 1320 nm, we evaluate the detector performance under various signal encoding schemes and observe data reception at 100 Gbps.

## Results

**Device fabrication and material characterization.** The devices are fabricated on a conventional silicon-on-insulator (SOI) substrate using TASE, the process is illustrated in Fig. 1a. First, we pattern the top silicon layer by a combination of e-beam lithography and dry etching of silicon. The features of the future detectors and all silicon passives, such as waveguides and grating couplers are etched simultaneously in this step (Fig. 1a(1–2)), which provides for inherent self-alignment of the III–Vs and silicon features.

The silicon features are then embedded in a uniform $SiO_2$ layer, deposited as a combination of atomic layer deposited (ALD) and plasma-enhanced chemical vapor deposited (PECVD) $SiO_2$, which is thinned down and planarized by chemical-mechanical polishing (CMP). An opening in the oxide is made to expose the silicon in areas where the Si will be replaced with III–V material. The Si is then selectively etched back using tetramethylammonium hydroxide (TMAH) to form a hollow oxide template with a Si seed at one

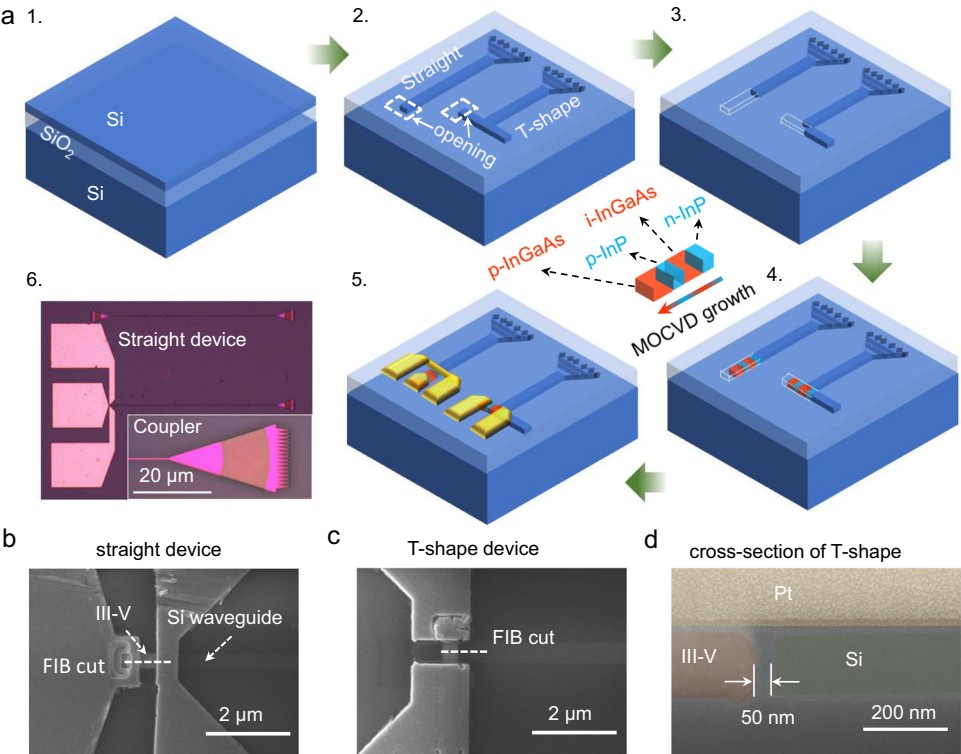

**Fig. 1 Template-assisted selective epitaxy fabrication process. a** Simplified schematic of the fabrication process: 1. SOI wafer. 2. Patterned top Si layer and openings (300 nm * 600 nm). 3. Partial etch-back of Si to form hollow SiO$_2$ template with a Si seed. 4. Metal-organic chemical vapor deposition (MOCVD) of heterostructure p-i-n device, the arrow in the inset shows the growth direction. 5. Ni/Au contacts on the nanostructure. 6. Optical microscope image of a straight device and the coupler in the inset. **b** Top view scanning electron microscope (SEM) image of a straight device, showing the focused ion beam (FIB) cut line. **c** Top view SEM of a T-shape device showing the FIB cut line. **d** Cross-section false-colored SEM image of a T-shape device showing the ~50 nm wide oxide-filled gap separating the Si waveguide and the III–V active material.

extremity (Fig. 1a 3). TMAH is an anisotropic wet etchant that results in a smooth but tilted Si (111) facet.

In the next step (Fig. 1a 4) the desired III–V profile is grown within the template by metal-organic chemical vapor deposition (MOCVD). In this work we grow a n-InP/i-InGaAs/p-InP/p-InGaAs sandwich structure. The role of the two wider bandgap InP regions is to improve carrier confinement in the i-InGaAs region. We believe that the presence of the heterostructure significantly improves performance compared to our earlier work with pure InGaAs[33]. Note that the InGaAs region is not completely intrinsic but will have a slight n-type doping as a result of parasitic carbon doping in the MOCVD. The p-InGaAs growth is intended to improve contacting as it can be difficult to obtain a good Ohmic contact on p-type InP. Details on the growth conditions can be found in Supplementary Note 2 and Supplementary Table 1. Following growth, the top oxide is uniformly thinned further by reactive ion etching (RIE) to obtain a thickness of ~10 nm to enable the thermal measurements and facilitate contacting. Top-view SEM and energy dispersive x-ray spectroscopy (EDS) analyses are used to distinguish the area from the n-InP/i-InGaAs/p-InP/p-InGaAs sandwich structure and Ni-Au metal contacts are implemented by e-beam lithography and lift-off (Fig. 1a 5).

A series of devices were fabricated on a SOI wafer with a thickness of 220 nm, and varying device width ($W_{PD}$) of the III–V region with $W_{PD}$ = 200 nm, $W_{PD}$ = 350 nm, and $W_{PD}$ = 500 nm. At the opposite end of the waveguide, a focusing grating coupler is implemented for diffracting light at around 1320 nm for a targeted incident angle of 10°. As it is illustrated in Fig. 1a, we focused on two different device architectures. One, where the III–V material is grown as an extension of the waveguide (Fig. 1b) – we refer to this

as the "straight" device. In addition, there is a structure where the III–V material is grown with a nucleation seed on a separate Si structure orthogonal to the optical waveguide with the grating coupler – we refer to this as the "T-shape" device (Fig. 1c). Figure 1d shows the cross-section SEM of a T-shape device with the focused ion beam (FIB) cutting line shown in Fig. 1c: the Si waveguide is separated from the orthogonally grown III–V structure by a small oxide-filled gap (~50 nm).

The motivation behind the two device architectures is that each structure provides different benefits and challenges. The straight structure is the easiest from a conceptual point of view and the propagating mode will be coupled directly from the silicon waveguide to the III–V region. The drawback is that contacts will inevitably need to be in the path of the optical mode which will most likely result in a higher absorption loss from the metal, and if there are localized defects at the Si/III–V interface these will also be in the path of the propagating light. In the T-shape structure, the III–V is grown orthogonal to the waveguide, therefore no contacts need to be placed on top of the waveguide and the waveguide may directly end at the i-InGaAs region. Any defects associated with the Si/III–V interface will also be no longer in the optical path. However, the coupling efficiency across the thin gap is unknown and this might lead to back-reflections in the silicon waveguide. In this proof of concept work we demonstrate the feasibility of such a coupling scheme, but we expect significant coupling losses. These could most likely be improved by a more sophisticated waveguide design. Conceptually there are advantages and challenges to both approaches, so investigating this trade-off was one of the objectives of the present work.

To investigate the device architecture and material quality, a TEM lamella was prepared on a straight device using FIB, with

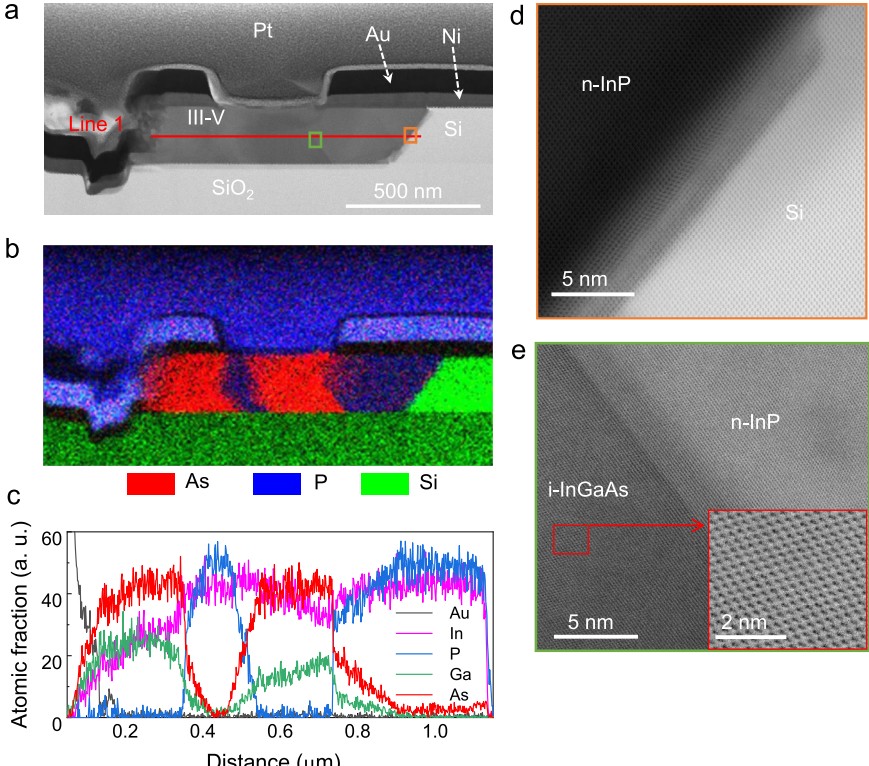

**Fig. 2 Nanostructure of p-i-n device and EDS analysis. a** Bright field scanning transmission electron microscopy (STEM) cross-section overview image of a 350 nm straight device. **b** Energy dispersive spectroscopy (EDS) overview map of the p-i-n device. Here, P, Pt, and Au appear blue because the EDS signal of P K-shell is near to the Pt and Au M-shell. **c** EDS elemental profiles acquired along Line 1 defined in **a**. **d** High-resolution bright field scanning transmission electron microscopy (STEM) images taken at Si/n-InP interface (orange box marked in **a**). **e** High-resolution bright field STEM images taken at n-InP/i-InGaAs interface (green box marked in **a**), inset: high-resolution STEM of the i-InGaAs.

the FIB cutting line shown in Fig. 1b. Figure 2a shows the overview STEM image of the sample. A tilted Si (111) crystal facet is observed at the III-V/Si interface where the growth initiated. This is due to the anisotropic TMAH etch, which will terminate on a (111) plane.

EDS was performed along the device, as presented in Fig. 2b. From left to right we can identify following regions: p-InGaAs (in red), p-InP (in blue), i-InGaAs (in red), and n-InP (in blue). The observed shape and width of the sections stem from the individual crystal facets formed during the growth sequence which can lead to a device-to-device variability. Further studies to finetune the epitaxial processes are ongoing. We note that using a similar geometry on a InP substrate, the growth of sharp and vertical quantum wells was demonstrated[34,35].

Figure 2c presents a line profile acquired along the "Line 1" as indicated in Fig. 2a. Pronounced transition regions with apparently graded compositions are visible and attributed to the non-orthogonal alignment of the crystal growth facets to the beam direction, with the exception of the starting Si/InP interface. We also observe that the intrinsic InGaAs region appears more In-rich (75 % In) compared to the p-region (53% In), which can be explained by the effect of the introduced doping precursor on composition and growth[22,36].

The high-resolution (HR) bright field (BF) STEM image in Fig. 2d shows an overall sharp Si/InP interface, with projection effects appearing as blurred area. Similarly, the HR BF-STEM image in Fig. 2e shows a sharp interface between the i-InGaAs and the n-InP with some projection effects. The inset shows a representative HR BF-STEM of the i-InGaAs with high crystalline quality which enables electrically stimulated light emission from

the device. No dislocations are observed in this or in other similar cross-sections, whereas we do observe regions with stacking faults.

**Thermal effects during device operation.** To reach a thorough understanding of the thermal effects on nanometer scale and hence be able to understand its impact on device performance, we performed thermal simulations using ANSYS Parametric Design Language (APDL) and experimental measurements using SThM[37] on a T-shape device ($W_{PD}$ = 500 nm). The SThM allows to measure the surface temperature quantitatively with about 10 nm lateral resolution while applying a bias to the device. Specifics of the SThM setup as well as the other setups used can be found in Supplementary Note 1.

Figure 3a shows the simulated temperature increase as a function of applied forward bias (LED operation) along with a measured SThM result. The resulting temperature rise compares well with SThM data, where an AC modulated bias of about 3 V amplitude is applied on the device. Figure 3b, c shows the simulated and measured temperature distribution of the device operating at 3 V, from which we observe a very good agreement between experimental and simulation data in terms of temperature increase. Figure 3d shows the temperature increase along the black and red dashed line in Fig. 3c. Except for a local high-temperature region at the contact edge, which might be caused by local high resistance at the contact, the overall temperature increase in the III–V is only around 15 K.

The temperature increase of the device while working as detector in reverse bias depends on the light injection in the III–V region. In addition to creating an electron-hole pair, the

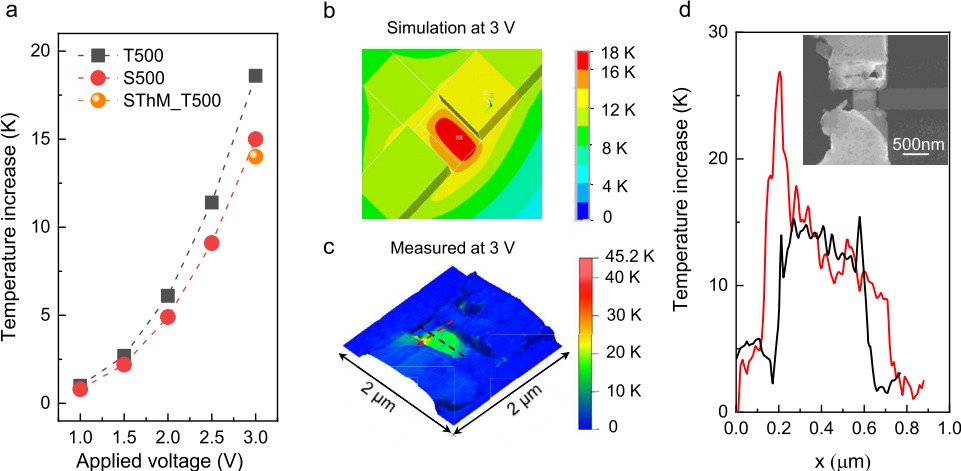

**Fig. 3 Thermal characterization of a 500-nm T-shape device. a** Temperature increase dependent on applied voltage (operating as emitter), the black squares show the simulation results of a T-shape device with device width of 500 nm (T500), the red dots show the simulation results of a straight device with device width of 500 nm (S500), the orange dot shows the scanning thermal microscopy (SThM) measured result on T500. **b** Simulated temperature distribution of the 500-nm T-shape device operating at 3 V. **c** SThM image showing topography and color-coded temperature rise of the device operating at 3 V. **d** Temperature profile from SThM results along the black and red dashed line in **c**, inset: scanning electron microscope (SEM) image of the measured device.

absorption of a photon will also lead to the creation of phonons and heating of the device. While light injection is not possible in the SThM setup, we can measure any temperature increase associated with the reverse bias drift current, but we expect this to be negligible in comparison. Therefore, we performed thermal simulations in the reverse bias case and the result shows a temperature increase of ~50 K when assuming a light injection of 3.16 mW (corresponding to the maximum laser power used for detection). Hence, we conclude that we do not expect any catastrophic thermal breakdown in these devices under the used measurement conditions, which is in agreement with our experimental observations. The detailed simulation results of the detector can be found in Supplementary Fig. 4.

**Electroluminescence as emitter**. In forward bias the device can be used as an LED, where the undoped InGaAs region acts as the active region for the emitter. However, the grating couplers cannot be used in this mode as their transmission is optimized for 1320 nm while the emission from the InGaAs region is centered at 1550 nm. The choice of wavelength of the grating couplers was motivated by the availability of existing designs developed for on-chip photodetectors for data communication applications, which were centered around 1320 nm[8]. Emission measurements are therefore performed in a free-space coupled optical setup in reflection mode. More details on the electrical/optical measurements can be found in Supplementary Note 3.

The electroluminescence (EL) measurements were performed under continuous wave (CW) operation from 80 K to 300 K. Figure 4a shows the EL spectra of the T-shape device with $W_{PD} = 350$ nm (T3). An EL peak centered at 1550 nm is observed at room temperature when a forward bias of 2.5 V is applied on the device. As illustrated in Fig. 4a, a blueshift of the EL peak is observed upon increasing the applied bias on the device. The reason for the different biasing regimes at different temperatures is the temperature-dependent threshold voltage shift, which is illustrated in Fig. 4b. A significant forward current increase with temperature is observed when comparing the same bias voltage, which results in the EL intensity increase as the temperature rises. The reverse current stays constant as temperature increases from 80 K to 150 K and then increases as the temperature increases from 200 K to 300 K. This is likely due to the activation of defect

centers acting as current paths. When the temperature increases from 80 K to 150 K, the defects are "frozen" and the reverse current stays constant. As the temperature further increases, the defects start to be activated and we see a reverse current increase from 200 K to 300 K. Additionally, the voltage related to the lowest current shifted from 0 V to −1 V at 250 K. This voltage shift is correlated to the trapped carriers within the structure which induce extra current.

For detailed comparison, the EL peak wavelength dependence on injection current at various temperatures from 80 K to 300 K is shown in Fig. 4c. The injection-current dependent blueshift of the EL peak is likely due to the band filling effect of the carrier injection in the active region. The injected carriers prefer to first fill the states with lower energy and emit light with longer wavelengths. As the injection current or bias voltage increases, the states with lower energy are occupied and the carriers fill the higher energy states, which results in EL with shorter wavelengths. Increased carrier injection is also predicted to influence the refractive index (plasma dispersion effect) which was observed in our microdisk lasers[38], but as we here consider an LED without a resonant cavity, the impact of a change in refractive index should be minimal. In addition to the EL blueshift, a temperature-dependent redshift of the EL peak was observed when comparing the same injection current, which is due to the bandgap shrinkage of InGaAs as the temperature increases. This result correlates with the thermal measurements, where we observed a 15 K temperature increase at 3 V, indicating that device self-heating was not a limiting factor for the LED operation.

**Responsivity as photodetector**. In order to evaluate the detector performance of the devices, we measured them in a fiber-coupled setup. For the dynamic measurements, light is coupled from a single-mode optical fiber into the silicon waveguide via the grating coupler. More details on the transmission characteristics of the grating coupler can be found in Supplementary Fig. 6, where also the coupling efficiency and waveguide losses are presented. As the transmission spectrum of the grating coupler is quite narrow, all the dynamic detection measurements are performed at a wavelength of 1320 nm. Our earlier work on smaller form factor detectors[33] showed a non-linear spectral dependence,

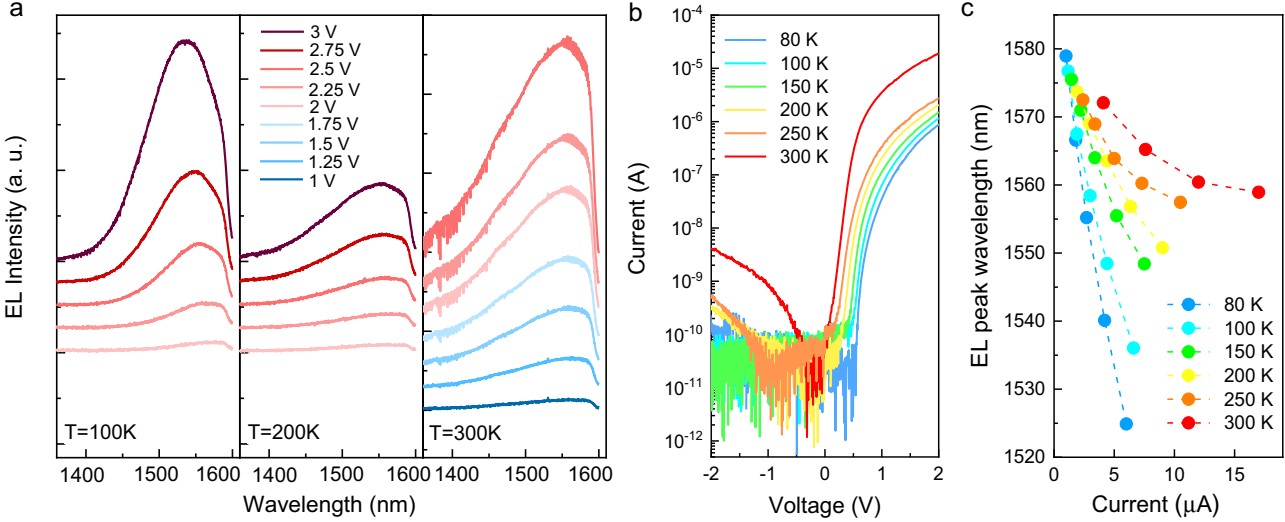

**Fig. 4 Electrically pumped light emission. a** Electroluminescence (EL) spectra of a 350 nm wide T-shape device (T3) under continuous wave (CW) operation, measured at 100 K, 200 K, and 300 K. **b** Current–voltage (*I–V*) curves measured from 80 K to 300 K. **c** EL peak wavelength dependence on injection current at various temperatures from 80 K to 300 K. EL spectra and peak energy plots in photon energy can be found in Supplementary Fig. 5.

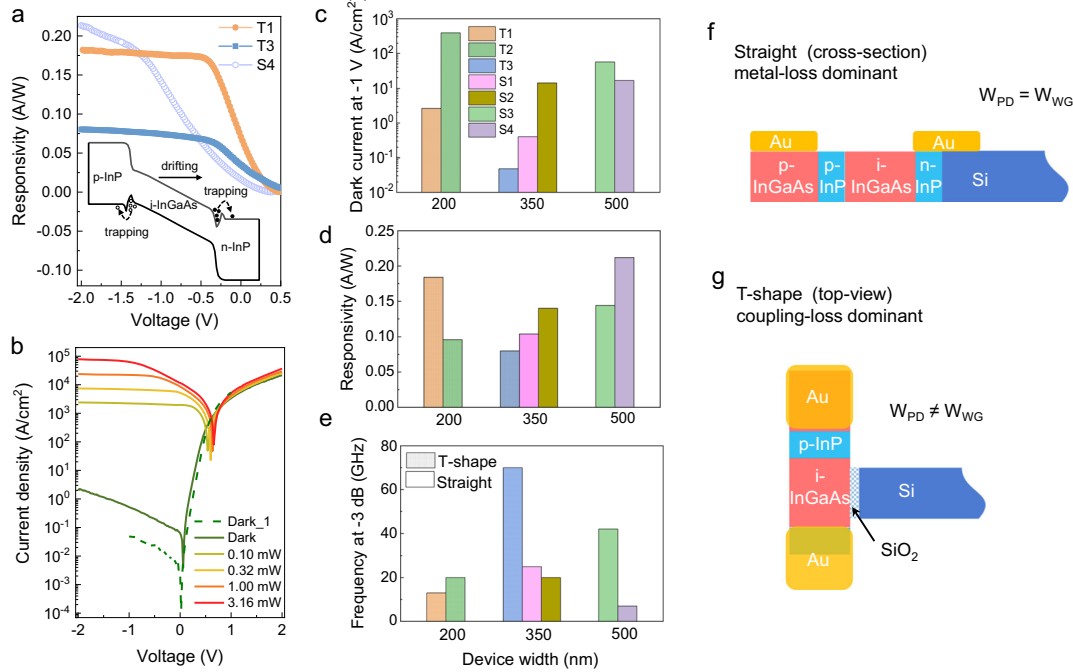

**Fig. 5 Static characterization of the p-i-n photodetectors. a** Responsivity of T1, T3 (T-shape), and S4 (straight), inset: energy band diagram of the p-i-n device. **b** *I–V* curves of T3 without light (dark green) and with 1320 nm waveguide coupled laser power varying from 0.1 mW to 3.16 mW. **c** Dark current (at −1 V) dependence on device architecture and device width. **d** Responsivity (at −2 V) dependence on device architecture and device width. **e** $f_{3dB}$ dependence on device architecture and device width. **f** Schematic of the cross-section of a straight device, $W_{PD}$ is the device width and $W_{WG}$ is the Si waveguide width. **g** Schematic of the top-view of a T-shape device.

therefore, we evaluated this for these waveguide-coupled structures as well. As expected, we did not observe any unusual trends. More information on the spectral and free-space power dependence can be found in Supplementary Figs. 7 and 8.

We investigated waveguide coupled devices with different architectures and dimensions. Figure 5a shows the responsivity of two T-shape (T1, T3) and one straight (S4) devices excluding the 6 dB coupling loss of the waveguide and the coupler. As we increase the reverse bias, the responsivity of all devices increases with the reverse voltage in two slopes which can be described by the different carrier transport mechanisms: carrier trapping at the

heterojunction interfaces and carrier drifting in the intrinsic region of the junction, as shown in the inset of Fig. 5a. Different slope values and turning points for the two slopes can be observed on devices with various band offsets and depletion layer lengths formed during MOCVD growth. T1, T3, and S4 show responsivities of 0.18, 0.08, and 0.21 A/W at −2 V, respectively. This corresponds to an external quantum efficiency ($\eta_{EQE}$) of 19%, 8%, and 22% ($\eta_{EQE} = \frac{R}{\lambda} \times \frac{hc}{e}$, where $R$ is the responsivity, $\lambda$ is the wavelength, $\hbar$ is the Planck constant divided by $2\pi$, $c$ is the speed of light in vacuum, $e$ is the elementary charge). We also expect

there to be significant additional absorption losses resulting either from the metal contacts placed directly on top of the III–V region (straight shape) or from the coupling from the Si waveguide to the III–V absorption region (T-shape), hence the values of responsivity presented herein should be considered as a lower boundary. Electromagnetic simulations were performed to obtain theoretical values of the responsivity (see Supplementary Note 4). The simulation parameters and results of the straight and T-shape devices are described in more detail in Supplementary Table 2 and Supplementary Fig. 9. Depending on the geometry, the amount of light absorbed in the i-region lies between 10% and 20% which is in good agreement with the measured values. The simulation results show higher metal loss and slightly more absorption in the p-InGaAs for the straight device. The reason for the slightly different scaling behavior is that for the straight devices increasing the width of the detector also means increasing the width of the waveguide. This will impact the mode distribution and the current density. Whereas in the T-shape photodetectors the width of the detector is independent of the waveguide dimensions, so increasing the width of the photo-detector will increase the length of the region where the light impinging from the waveguide is absorbed.

Figure 5b shows the current-voltage ($I$–$V$) curves without light (dark green) and under illumination with a 1320 nm laser coupled from the Si waveguide. The dashed green $I$–$V$ curve was measured at a probe station with high resolution, displaying a dark current of around 0.048 A/cm$^2$ at $-1$ V (normalized to the device cross section), which is two decades lower compared to our previously reported pure InGaAs devices[33] and comparable to high-speed bonded membrane III-V photodetectors[8]. We believe this improvement stems from the use of the double heterostructure.

Figure 5c, d, e show the normalized dark current at –1 V, the responsivity at $-2$ V and the $f_{3dB}$, respectively. By comparing these parameters, a clear inverse trend for the responsivity and the $f_{3dB}$ is observed for each device, i.e. those devices showing a higher responsivity tend to register a lower $f_{3dB}$, if we compare the trend within the same device width. We believe this trade-off is mainly due to a longer time required to extract the carriers for a device with higher responsivity. For example, if contacts are (unintentionally) positioned to absorb a significant fraction of light, it will result in a smaller responsivity as the light absorbed by the contacts is lost, but it might also make the device faster if the contacts absorb the light generated at the edges of the i-region.

Figure 5f, g show the schematics of the straight and T-shape device. In the straight device, the Si waveguide serves also as the Si seed for the MOCVD growth of the p-i-n structure and in this case the photodetector width is always equal to the waveguide width ($W_{WG}$). For the dimensions we are looking at in this structure, the mode should be well confined in the Si waveguide, hence a change of the detector/waveguide width should not directly influence the measured absolute current. However, it would change the current density as this would be calculated over a larger cross-section area. In this structure, we would possibly expect an increased absorption if instead we vary the length of the i-region, but as this is determined by the duration of epitaxial growth it can not be varied on a device-to-device level. A significant metal loss is expected in this structure as the light is absorbed by the Au contact on top of the waveguide.

In the T-shape structures, the propagating light impinges on the i-region orthogonally from the Si waveguide, in this case the detector width is independent of the waveguide width. If we make the detector wider, we may expect that more light will be absorbed, so that we should see an increase of absolute current, and possibly also in current density – depending on the magnitude of such an increase. Another advantage is that the

Au contacts as well as any inherent defects at the Si/III-V are not directly in the line of the light. However, this device is most likely limited by the coupling from the Si waveguide to the III-V region over the SiO$_2$ gap.

**High speed operation**. To achieve high data rates and a high signal-to-noise ratio (SNR), not only the cut-off frequency but also the saturation photocurrent is of interest. In addition to the highest current, higher modulation formats, such as the 4-level pulse-amplitude modulation demonstrated here, also have increased requirements for linearity. Following the discussion in Williams and Esman[39] we identify three components that limit linearity: thermal effects, voltage drop in the series resistance, and carrier screening.

High optical and electrical power densities ultimately cause catastrophic thermal failure. We concluded from our thermal characterization and simulation results that a reverse voltage of –2 V together with an optical power level of 7 dBm was usually safe in this regard. The other two effects mentioned above also limit the linearity, but do not cause catastrophic failure. High photocurrents lead to a voltage drop in the series resistance and therefore reduce the reverse voltage applied across the p-i-n junction. An estimation of the maximum photocurrent can therefore be made as:

$$I_{sat} = \frac{V_{rev} + V_{bi}}{R_s} \tag{1}$$

In addition, photocarriers in the intrinsic region form a screening field proportional to the optical power[40], which also limits the optical power. Since this screening field depends on the excess carrier density, the effect is expected to be more pronounced in small detectors.

To investigate the impact of these effects, we measured the linearity of a 200 nm wide device (T2) at different bias voltages and small-signal frequencies. Figure 6a, b show the resulting linearity curves. The measurements were fitted with the saturation expression

$$P_{out} = \frac{\left(\alpha P_{in}\right)^2}{1 + \left(\frac{P_{in}}{P_{sat}}\right)^2}, \tag{2}$$

where $\alpha$ contains the responsivity $R$ as well as system RF losses and the 50 $\Omega$ load resistance. The saturation power value $P_{sat}$ is the 3 dB compression point and is given in the figure legend. A clear bias dependence can be seen, whereas no clear frequency dependence was observed. The linearity measurements suggest that under safe operating conditions, an input power of up to 10 dBm still results in a fairly linear response. Due to the scaling of the power limitations we expect wider devices to perform even better.

Figure 6c shows a bandwidth measurement of device T3 ($W_{PD} = 350$ nm) corrected for the system losses, the same device as measured for the emission. Consistent with the DC responsivity, the RF power at zero bias is 12 dB lower than the saturated value at $-1$ V. At $-1.5$ V, the device shows no clear cut-off up to the setup-limited frequency of 70 GHz. The ripples in the frequency response are most likely caused by RF reflections at the unterminated photodiode. In the present devices contacts have not been optimized for RF performance, we believe that optimization of device and contact design could lead to further gains in performance, in line with what was observed from detectors based on wafer bonding[8].

We performed a data transmission experiment on the same device to show the capability of the fabricated photodiodes. Figure 6d, e shows the digitally interpolated eye diagram of 50 Gbit/s on-off keying (OOK) and 100 Gbit/s four-level pulse-amplitude modulation

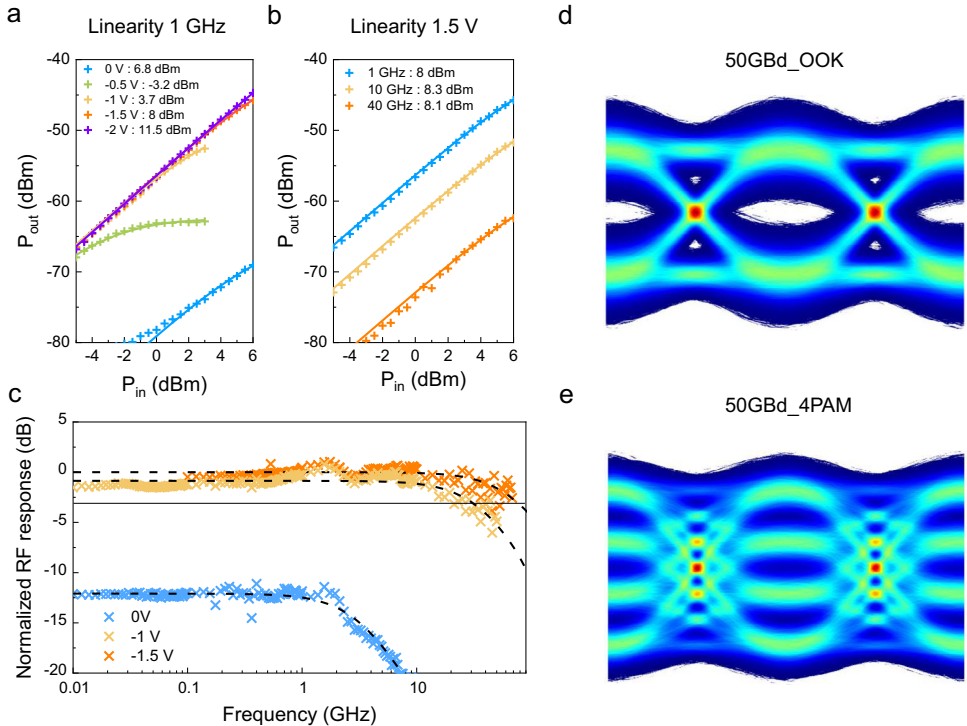

**Fig. 6 Linearity and dynamic characterization of the photodetectors. a** Linearity at 1 GHz measured on device T2 at different reverse bias voltages, showing output power ($P_{out}$) vs input power ($P_{in}$). **b** Linearity at a constant 1.5 V reverse bias measured on device T2 for different small-signal frequencies. **c** Radio frequency (RF) signal response dependence on modulation frequency of the input signal measured on device T3. **d** 50 GBd OOK eye diagram measured on device T3. **e** 50 GBd 4PAM eye diagram measured on T3.

(4PAM) measured on T3. We use a non-return to zero (NRZ) signaling scheme for both rates. For the 50 Gbit/s transmission we achieve a bit-error rate (BER) of $3.21 \times 10^{-5}$ which is below the hard decision forward-error correction (FEC) limit of $3.8 \times 10^{-3}$ (ref. [41]). In contrast, we achieve a BER of $1.17 \times 10^{-2}$ for the 100 Gbit/s transmission. This BER is below the soft decision FEC limit of $4.2 \times 10^{-2}$ (ref. [42]). A subset of the data-transmission results of a single device have been presented at the Optical Fiber Communication (OFC) conference[43].

Table 1 shows the performance metrics of other state-of-the-art III–V on Si near-infrared photodetectors compared with the performance of the detectors shown in this work. For our own work we include the highest speed device, this device has a lower responsivity than some of the others we measured. For non-waveguide coupled devices the responsivity is a calculated value based on various estimates.

## Discussion

In conclusion, we demonstrated waveguide coupled III–V heterostructure photodiodes monolithically integrated on Si with sub-micron dimensions. The devices show light emission centered at 1550 nm when operating in forward bias as an LED. A blueshift was observed with increasing bias which we attribute to band-filling effect, and the threshold voltage of the diodes also showed a strong temperature dependence.

In photodetection mode the devices show a dark current down to 0.048 A/cm² at –1 V and a responsivity up to 0.2 A/W at –2 V. This value is not corrected with respect to additional losses due to coupling from the Si waveguide to the III-V active region and should be understood as a lower boundary. With the grating couplers centered around 1320 nm, high-speed detection with bandwidth exceeding 70 GHz was demonstrated, which enables data transmission at 50 GBd with OOK and 4PAM. A trade-off

was observed among different devices in terms of responsivity and $f_{3dB}$. We believe that there is significant potential to further optimize the device width, the length of the i-region, or improve the coupling from Si to III–V for the T-shape devices and contacting scheme for straight devices, which could lead to further improvement in responsivity and detection beyond 100 Gbps. Due to the many different trade-offs in terms of contact placement, coupling losses and different geometrical dependencies, a one-to-one comparison among straight and T-Shape devices is not possible in this work. However, the T-shape device provides much more freedom in terms of design and possibilities for further performance optimization. Therefore, we believe this is the most desirable architecture for future work.

Thermal effects were evaluated by simulation and SThM for both emission and detection operation and in both cases we find an acceptable temperature increase for stable operation. These findings also correlate with the optical measurements and the measured device linearity at high frequencies.

The presented in-plane integration of the III–V heterostructure p-i-n diode self-aligned to a Si waveguide represents a new paradigm for mass production of densely integrated hybrid III–V/ Si photonics schemes. By using the same approach for the integration of the detector and the emitter and an integration technique that enables heterojunctions along the growth direction, this approach can also be extended to an all-optical high-speed link on Si without the need for evanescent coupling. Compared to previous demonstrations in Table 1, we do not rely on pick-and-place methods, multi-level coupling, regrowth or diffusion for integration of doping profiles. We can leverage the self-alignment with nm precision of passive and active components and the in-situ growth of heterojunctions. On the same chip we implemented optically pumped photonic crystal (PhC) emitters covering the entire telecom band[44]. The coupling from the 1D PhC emitters to the waveguide coupled photodetectors demonstrated

**Table 1 Performance metrics of state-of-the-art III–V on Si near-infrared photodetectors.**

| Material | Substrate | Device integration | Waveguide coupled | R [A/W] | $f_{3dB}$ [GHz] | Dark current | LED $\lambda$[nm] | Refs. |
|---|---|---|---|---|---|---|---|---|
| InGaAs | Si (111) | MOCVD | Polymer | 0.68 @ 850 nm | 3.8 | 45 nA @ −1 V | 1100 | 18 |
| p-i-n InP/InAsP | Si | Pick & place | PhC | - | 10 | 20 nA @ −1 V | 1260 | 19 |
| InGaAs | Si (111) | MOCVD NW array | No | 0.25 @ 635 nm[a] / 0.003 @ 1550 nm[a] | - | 0.2 mA/cm$^2$ @ −1 V | - | 47 |
| InGaAs | SOI | Wafer bonding | Si | 0.4 @ 1260 nm[a] | 65 | 0.033 mA/cm$^2$ / 0.2 nA @ −4 V | - | 8 |
| InGaAs | SOI | MOCVD TASE | No | 0.68 @ 1346 nm[a] | 25 | 2.8-7.5 A/cm$^2$ / 1.7-36 nA @ −2 V | 1600 | 33 |
| InP/ InGaAs QW | SOI | MOCVD TASE | No | 0.2 @ 1550 nm[a] / 0.6 @ 1310 nm[a] | 40 | 0.12 nA @ −1 V | - | 48 |
| p-i-n InP/ InGaAs/InP | SOI | MOCVD TASE | Si | 0.08 @ 1320 nm | 70 | 0.048 A/cm$^2$ / 0.04 nA @ −1 V | 1550 | This work |

[a]calculated responsivity.

here should be straightforward as they are implemented in the same plane and integrated in the same MOCVD run. What remains is the optimization towards electrically actuated lasing. This is naturally far from trivial, but we believe it might be possible to achieve this in the future based on in-plane epitaxial growth provided by TASE.

## Methods

**Device fabrication**. First, a conventional SOI substrate with top silicon thickness of 220 nm is prepared which defines the thickness of the III–V device (Fig. 1a 1). Then, the top silicon layer was patterned by a combination of e-beam lithography using HSQ resist and HBr dry etching of silicon, forming the features of the future waveguides and grating couplers (Fig. 1a 2). The silicon features were then embedded in a uniform SiO$_2$ layer which in the following steps serves as the oxide template for the III–V growth. An opening was then made to expose the silicon where the selective back-etching using TMAH starts, exposing at one extremity a silicon seed (Fig. 1a 3). In the next step, the III–V profile was grown within the template by MOCVD (Fig. 1a 4). Following the growth, the top oxide was etched further down and the metal contacts were implemented by sputtering and evaporation of Ni-Au metal (Fig. 1a 5).

**Material characterization**. To investigate the device architecture and evaluate the material quality, a TEM lamella was prepared using an FEI Helios Nanolab 450S FIB. The cut was conducted along the growth direction on a 350 nm straight-type device. The lamella was then investigated by STEM with a double spherical aberration-corrected JEOL JEM-ARM200F microscope operated at 200 kV, which permitted to assess the crystalline quality of the various III–V regions and the Si seed. Using a liquid-nitrogen-free silicon drift EDS detector, element mapping and species quantification were carried out with the commercial Gatan Micrograph Suite ® (GMS 3) software by assuming the lamella thickness of ~100 nm and using the theoretical k-factors for the quantification.

**Thermal characterization**. Thermal effects of the photodiodes were investigated by scanning thermal microscopy (SThM), which is performed in a high-vacuum (<10$^{-6}$ mbar) chamber at room temperature in the Noise-free labs at IBM Research Europe - Zurich[45]. The SThM-based technique relies on a micro-cantilever with integrated resistive sensor coupled to the silicon tip, which enables temperature measurements with down to few nanometer spatial resolution and sub-10 mK temperature resolution[46]. The temperature of the tip $T = 267$ °C is known by detecting the lever voltage and was calibrated before the scan when the tip is out of contact. The scan is operated in contact mode whereby the contact force is monitored and controlled by a laser deflection system. The temperature of the sample was modulated by applying an AC voltage with frequency of $f = 1$ kHz on the device. A series resistance of 10 kΩ was used during the measurement. The local temperature on the sample is thermally coupled through the tip to a resistive sensor integrated in the silicon MEMS cantilever. The change of the sensor temperature leads to the change of the electrical resistance of the cantilever, which is tracked using a Wheatstone bridge circuit. Details on the setup can be found in Supplementary Fig. 1.

**Thermal simulation**. Thermal simulations were carried out using commercial finite element method with APDL, which uses Fourier's law of heat conduction to calculate the heat flow. When simulating device operation as emitter, a uniform heat generation was applied on the III–V region with a total heat power equal to the electrical power applied while doing the SThM measurement. In the simulation, the back side of the Si substrate is assumed to be at a constant temperature of 300 K. For device operation as detector, a Gaussian-distributed heat generation was applied on the III–V region, which simulates a laser spot with a spot size of 1 μm. Only heat conduction was accounted for in the simulation since heat convection and radiation can be neglected considering the measurement conditions and temperature increase.

**Electroluminescence spectroscopy**. The EL measurements were performed under CW operation from 80 K to 300 K. The device was placed in a cryostat with 4 probes and with capability of cooling down to 10 K. An Agilent 1500 A was used as both device parameter analyzer and power supply. The light emission is collected from the free-space by an objective with a magnification of ×100 and numerical aperture of 0.6 and detected by an InGaAs line array detector (Princeton Instruments, PyLoN-IR 1.7) which is combined with a grid diffraction spectrometer (Princeton SP-2500i). An integration time of 30 s was used to get high signal-to-noise ratio. The layout of the setup can be found in Supplementary Fig. 2.

**High-speed detection**. Responsivity and high-speed measurements were performed in an optical setup with an optical fiber. For bandwidth measurements, a CW tone was generated using a 70 GHz Keysight synthesizer and modulated onto a 1320-nm optical carrier. The system frequency response was calibrated using a commercial 67 GHz u2t photodiode. For the data transmission, an electrical data

signal was first generated using a Mircram 100 GSa/s digital to analog converter (DAC). The output of the DAC was then amplified using a SHF driver amplifier (DA) with 3 dB bandwidth of 55 GHz. A u2t Mach-Zehnder modulator was used to transfer the electrical signal to the 1320 nm optical carrier generated with a Keysight tunable laser source (TLS). In the next step, the optical signal was amplified to the optimum power using a FiberLabs Praseodymium doped fiber amplifier (PDFA). High-speed RF probes were used to extract the RF signal from the device under test (DUT) and a reverse bias voltage of –1.5 V was supplied to the DUT using a SHF bias tee. At the receiver, an Agilent 160 GSa/s digital storage oscilloscope (DSO) was used to record the generated RF signal. A 30 cm RF cable was used to connect the DSO to the bias tee, limiting the frequency response of the full system. An offline digital signal processing was used to process the recorded signal. The digital signal processing setup comprises of signal normalization, timing recovery, linear equalization, and non-linear equalization (see details in Supplementary Fig. 3).

## Data availability

Source data are provided with this paper and also available on Zenodo: https://zenodo.org/record/5643649. Source data are provided with this paper.

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

## Acknowledgements

This work has received funding from the European Union H2020 ERC Starting Grant project PLASMIC (Grant Agreement #678567) and H2020 MSCA IF project DATENE (Grant Agreement #844541). We also acknowledge Y. Baumgartner and M. Seifried for design of the grating couplers, as well as L. Czornomaz, A. Schenk and V. Georgiev for technical discussions. We gratefully acknowledge technical support for CMP from Daniele Caimi and we thank the Cleanroom Operations Team of the Binnig and Rohrer Nanotechnology Center (BRNC) for their help and support.

## Author contributions

S.M. designed the devices and waveguides, S.M., P.T., and P.W. fabricated the devices. S.M. and H.S. conducted the material growth. P.W. and M.S. prepared focused ion beam

lamella and performed material characterization. P.W. and B.G. performed the thermal analysis. P.W and M.Sc. performed the emission characterization. P.W., P.T., M.B., B.I.B., and J.L. performed the detection characterization and data discussion. P.W. and K.M. wrote the article with contribution from all authors. K.M. led the project.

## Competing interests

The authors declare no competing interests.
