## [Peer Review File · Nature Communications]

Waveguide coupled III-V photodiodes monolithically integrated on SiREVIEWER COMMENTS

Reviewer #1 (Remarks to the Author):

Comments to the Author

The authors successfully present a method to grow the n-InP/i-InGaAs/p-InP/p-InGaAs sandwich structure which can work as both a photodetector and a light emitting diode, integrating with Si photonic circuits. I suggest this manuscript can be accepted for publication in Nature Communications after the following questions were addressed:

Growing the n-InP/i-InGaAs/p-InP/p-InGaAs sandwich structure integrating with Si photonic circuits is a key point of this manuscript. First, lattice mismatch is remaining between Si and III-V semiconductors, how can InP and InGaAs selectively grow onto Si but not SiO₂? Second, how can the authors control the length, width, thickness, and doping level for InP and InGaAs? Because these parameters are significant for optoelectronic performance. Afterwards, how can the authors distinguish the area of n-InP and p-InGaAs from the n-InP/i-InGaAs/p-InP/p-InGaAs sandwich structure before depositing Ni/Au? In conclusion, more detailed information for growing III-V semiconductors onto Si should be illustrated.

In Fig. 1d, there is a gap of 50 nm between Si and InP. In Fig2d, there is no gap (or a very small gap) between Si and InP. How can the authors control the gap length between Si and InP? Does the gap length have an impact on optoelectronic performance?

In page 9, the authors mentioned that “the grating couplers cannot be used in this mode as their transmission is optimized for 1320 nm and the emission from the InGaAs region is centered at 1550 nm”. Why do the authors don't design grating couplers for C-band operation? Is InGaAs suitable for photodetector at C-band operation?

In “Electroluminescence as emitter” section, the authors characterized the tunable spectra induced by injection current and temperature, but quantum efficiency is a significant parameter for LED. Can the authors provide this result in the revised manuscript?

Energy band diagram that has influence on both quantum efficiency of LED and carriers extraction of photodetector. Can the authors provide the energy band diagram of this PIN diode in the revised manuscript?

The authors mentioned that “The reverse current also increases as the temperature increases, this is expected as the reverse saturation current of a diode is highly temperature dependent and may further increase due to the activation of defect centers acting as current paths”. First, the reverse current at 80 °C to 150 °C doesn't increase with temperature. Second, the tendency of reverse current at 250 °C and 300 °C is different. The authors should give more detailed illustration.

In page 11, the authors mentioned that “As the transmission spectrum of the grating coupler is quite narrow, all the dynamic detection measurements are performed at a wavelength of 1320 nm”. Transmission characteristics grating coupler should be added. Moreover, broadband spectral response is an important factor for a photodetector, can the authors provide spectral response of this

photodetector? Additionally, there is just one grating coupler in a device, how does the authors define optimal coupling efficiency of the grating coupler? The accurate coupling efficiency is important for responsivity.

In page 11, the authors mentioned that "As we increase the reverse bias, the responsivity of all devices increases with the reverse voltage in two slopes which can be described by the different carrier transport mechanisms". The detailed information of carrier transport mechanisms should be added.

In page 12, the authors mentioned that "T1, T3 and S4 show responsivities of 0.18, 0.08 and 0.21 A/W at -2 V". Does the device width lead to this performance difference? Device width has an impact on absorption section and RC constant. Is the optimal width of photodetector equal to the width of waveguide? The authors should explain how the device width leads to different optoelectronic performance?

In Fig. 5a, responsivity of T1 and T3 keeps a constant at reverse voltage from -2 to -0.5 V, but responsivity of S4 decreases with reverse voltage from -2 to -0.5 V. The reverse voltage influences the length of depletion layer and the efficiency of carrier extraction. Why the tendency of these three devices is different?

In Fig 5a, T-shape structure and S-shape structure have different ways of combining the waveguide and the detector (resulting in different optical power reaching the detector), and the width of absorption region of T1, T3 and S4 is also different (leading to different absorption rates, transport efficiency, and transport time). As we can see, the only variable between T1 and T3 is the device width. Please explain the phenomenon: why the larger the device width, the lower the responsivity? On the other hand, there are too many variables between S4 and other two devices to form an effective comparison. But it can be seen that the relationship between responsivity and bias voltage of S4 is quite different from that of T1 and T3. Please explain the influence of the two patterns on the above relationship.

In page 12, the authors gave the simulation results and mentioned that "the amount of light absorbed in the i-region lies between 10 % and 20 % which is in good agreement with the measured value". The author should give the simulation results of T-shape and S-shape respectively and explain their differences. In addition, the authors should summarize which structure is better, T-shape or S-shape, and tell readers the answer.

In Figure 5d-e, the authors showed the responsivity and bandwidth of T-shape and S-shape as a function of device width. An obvious rule is ignored. The responsivity of S-shape structure increases with the increase of the device width, but the bandwidth decreases. The above phenomenon can be explained: the increase in the width of the absorption region brings about a larger light absorption area (larger responsivity), but it prolongs the carrier transport time (smaller bandwidth). However, the above dependencies of T-shape structure are opposite to that of S-shape structure. Please explain the reason for the different relationship between the responsivity/bandwidth and device width of T-shape and S-shape structures.

In page 12, the authors mentioned that "those devices showing a higher responsivity tend to register a lower f_{3dB} ". However, S3 doesn't follow this rule. S3 has a high responsivity and high bandwidth at the same time. The authors need to explain this.

Reviewer #2 (Remarks to the Author):

An interesting approach has been shown by this paper with very good results on III-V photodiodes monolithically integrated on Si. One of the major problem for III-V growth on silicon is defect generation. In this paper, the author did not discuss in detail about this. Can the authors add one paragraph to discuss this issue please? Although good individual device has been demonstrates, how is the yield? Recently, very good quantum dot lasers have been demonstrated on silicon, can the authors discuss the possible route to add lasers on this platform please.

Reviewer #3 (Remarks to the Author):

The author present a photodiode by using a template assisted selective epitaxy and characterize it as a photo detector and a led. The manuscript provided enough data to show the good quality of the epitaxy, characterize the device as emitter and a photodetector. The performance of the detector is decent. However, the template assisted selective epitaxy method is reported before, so it's not consider a novelty in this manuscript. The device working as a LED doesn't show extra novelty nor significance of this device. In addition, the manuscript lacks of comments/comparisons of the performance of the detector and the LED to literature that made by other integration method. It's a good paper to elsewhere but might be not with high impact enough to be published in Nature Communication. Some other questions listed below:

1. It would be good the show clearly how the opening in the oxide is located and dimensioned that for the epitaxy.
2. In the static characterization of the photodetectors, there are summary charts for dark current, responsivity and frequency. However, I would say that the device design set is not completed. It's a mess for readers.
3. Page 14 : Fig. 4a and b show the resulting  Fig. 6a and b show the resulting

Dear Reviewers,

Thank you very much for the detailed revision and suggestions! We found them very supportive and helpful to strengthen our study and we revised our manuscript accordingly. The major changes are highlighted in red and minor changes are highlighted in blue in the Track mode in the accompanying manuscript. Below are our point-by-point responses to the comments and suggestions raised by the reviewers.

Reviewer #1:

Comments to the Author

The authors successfully present a method to grow the n-InP/i-InGaAs/p-InP/p-InGaAs sandwich structure which can work as both a photodetector and a light emitting diode, integrating with Si photonic circuits. I suggest this manuscript can be accepted for publication in Nature

Communications after the following questions were addressed:

Growing the n-InP/i-InGaAs/p-InP/p-InGaAs sandwich structure integrating with Si photonic circuits is a key point of this manuscript. First, lattice mismatch is remaining between Si and III-V semiconductors, how can InP and InGaAs selectively grow onto Si but not SiO₂?

The unavoidable lattice mismatch between Si and III-Vs must be relieved somehow. Here, two methods are combined to achieve this goal in an effective way to suppress the formation and propagation of dislocations into the device. First, we use a specific Si surface having smooth (111) planes on which lattice mismatch is relaxed by forming a so-called misfit dislocation network, where all the defects are bound to the interface. The second strategy is to reduce the interface area to the sub-micron scale. This guarantees that the resulting crystal is formed from a single nucleation point only, thus suppressing the occurrence of grain boundaries from merging crystallites. In this case, the defects will be limited to a few nm's of the interface. Selective growth is based on the differences in the surface energies between the two materials, Si and SiO₂ respectively, which can be exploited by careful tuning the growth parameters. One can arrive at a condition where nucleation only takes place on a crystalline seed (in this case Si) and not on the masking material (in this case SiO₂), this is possible in both MOCVD and MBE growth. Whereas MOCVD is particularly suited for our purpose because of the long surface diffusion of the involved molecules, which can easily diffuse into recessed structures, unlike MBE where the process window is much smaller. We have added paragraphs discussing the material growth in the introduction part.

Second, how can the authors control the length, width, thickness, and doping level for InP and InGaAs? Because these parameters are significant for optoelectronic performance.

Width and thickness are determined by the geometry of the pre-patterned Si layer, this is the core strength of TASE growth compared to other techniques, namely, that we use a hollow oxide template to guide the growth rather than rely exclusively on the tuning of growth parameters. The length of the III-V segments is controlled by the growth duration.

The III-V composition as well as the doping is controlled by the composition of the gaseous precursor flows in the epitaxy reactor. In principle this is no different from other epitaxial techniques. An increase of the ratio of a certain pre-cursor will result in an increase of the fraction of this species in the final composition. For conventional MOCVD and MBE growth the community has had decades of experience to optimize this, whereas TASE is a relatively new growth technique so we are still working on improving our processes in order to arrive at an exact composition, doping level and

control of growth rates and facet formation. More details on the growth part were added to the supplementary file.

Afterwards, how can the authors distinguish the area of n-InP and p-InGaAs from the n-InP/i-InGaAs/p-InP/p-InGaAs sandwich structure before depositing Ni/Au? In conclusion, more detailed information for growing III-V semiconductors onto Si should be illustrated.

InP and InGaAs can be distinguished from a top-view SEM image overlapped with EDS analysis, as illustrated in figure 1. The EDS signals are collected only in the colored regions. Dopants cannot be resolved by EDS as the number of dopants is small relative to the total number of atoms. However, in initial work we calibrated doping on uniformly doped III-V rods grown in similar geometries and established a reference.

Figure 1. Top-view overlap of SEM and EDS of an as-grown p-i-n junction (a) Straight device (b) and T-shape device (c).

More detailed information on the selective growth of compound semiconductors on Si can be found in papers from our group [1] as well as articles from other researchers [2, 3].

We appreciate the reviewer's comments and have added more description on the growth process in the introduction part of the revised manuscript.

In Fig. 1d, there is a gap of 50 nm between Si and InP. In Fig2d, there is no gap (or a very small gap) between Si and InP. How can the authors control the gap length between Si and InP? Does the gap length have an impact on optoelectronic performance?

Thank you very much for the question. Fig. 1d and Fig. 2d show two different types of devices. Fig. 1d shows the cross-section of a T-shape cutting from the position shown in Fig. 1c. For this, we are showing the gap between the III-V and the Si waveguide. The gap length can be controlled by design of the electron beam mask layout. Please note however, that in TASE the III-V and Si regions are defined in the same lithographic step, so there is no inaccuracy stemming from overlay precision or alignment and gap-size is uniquely determined by the resolution of the process. The gap width has an impact on the light coupling efficiency from the Si waveguide to the III-V region. Fig. 2a and d show the cross-section of a straight device which is cut along the MOCVD growth direction, so the III-V / Si interfaces in Fig. 2 are the interface between the III-V and Si seed. In this case there is no gap as the III-V grows from the Si crystal. We have added an FIB cut line in Fig. 1b and discussion to make this more clear in the revised manuscript.

In page 9, the authors mentioned that "the grating couplers cannot be used in this mode as their transmission is optimized for 1320 nm and the emission from the InGaAs region is centered at

1550 nm”. Why do the authors don’t design grating couplers for C-band operation? Is InGaAs suitable for photodetector at C-band operation?

The grating coupler designs used here were borrowed from another activity of our group working on high-speed bonded III-V detectors optimized for datacom, which is integrated on top of a Si photonics platform [4].

We could have made new grating coupler designs which were suitable for 1550 nm, but our in-house detector characterization set-up is made for 1320 nm, so we would not have been able to test our devices there. This is a practical concern and not an inherent limitation of the technology. As illustrated by light emission at 1550 nm, as well as by the spectral dependence plots of the detectors provided in the supplementary material, the devices can also detect in this range. We have added corresponding information on the grating coupler in the revised manuscript.

In “Electroluminescence as emitter” section, the authors characterized the tunable spectra induced by injection current and temperature, but quantum efficiency is a significant parameter for LED. Can the authors provide this result in the revised manuscript?

Many thanks for the reviewer’s question!

Ideally, we would have liked to evaluate the quantum efficiency of the LED, however, this is not possible in the current set-up without gross assumptions. For an absolute external quantum efficiency (EQE), we would need the value of the electroluminescence (EL) intensity in W and the injection current in A. As the area of the device working as a LED is quite small (350 nm * 200 nm * 300 nm), the EL intensity is too weak to be measured with a powermeter. Also a conversion from injected power to detector counts is not straightforward for such relatively low luminescence devices. Potentially, we can obtain a normalized EQE by comparing the integrated EL spectrum under various current level, and we analyzed our data and obtained the normalized EQE shown below:

However, with this normalized EQE we cannot obtain a quantum efficiency value to show how good the performance of the LED is compared to the other LEDs shown in literature. Therefore, we prefer not to include this data.

We also tried to analyze the internal quantum efficiency (IQE) by using the ABC model [5, 6], however, we are not able to do this since the variation of the current level is not large enough to calculate both the A and C factors. The EL spectra we show in the manuscript span from 1 V to 2.5 V at room temperature, this corresponds to the current density from around 5000 A/cm² to 50000 A/cm² which is already high compared to the current level in literatures. If we go to higher current levels, the devices will be destroyed. Furthermore, as there is no resonant cavity, devices are emitting in all directions and we are only collecting a small fraction of the emitted light. If we go to lower current levels, the spectrometer will not be able to detect the spectra.

We are not expecting a high efficiency for the LED performance, since the structure is not designed or optimized for light emission. We measured and show the EL spectra in support of the device characterization. However, we are very interested in optimizing the device towards efficient detector and emitter in our future work.

Energy band diagram that has influence on both quantum efficiency of LED and carriers extraction of photodetector. Can the authors provide the energy band diagram of this PIN diode in the revised manuscript?

Many thanks for the reviewer's suggestion! We added the energy band diagram of the device in Fig. 5(a) as an inset illustration. This band diagram helps understanding the two slopes of the responsivity in Fig. 5(a).

The authors mentioned that "The reverse current also increases as the temperature increases, this is expected as the reverse saturation current of a diode is highly temperature dependent and may further increase due to the activation of defect centers acting as current paths". First, the reverse current at 80 °C to 150 °C doesn't increase with temperature. Second, the tendency of reverse current at 250 °C and 300 °C is different. The authors should give more detailed illustration.

We apologize for the incomplete description on the temperature dependent IV curves. As the reviewer correctly pointed out, the reverse current does not change from 80 K to 150 K while both the reverse current and forward current increase as the temperature increases from 150 K to 300 K. The reverse current increase with temperature after 150 K is likely due to the activation of defect centers acting as current paths. The temperature is related to the activation energy of the defects. The tendency of the reverse current at 250 K is different from others and shows the lowest current value not at 0 V but at around -1 V. In theory, the voltage for the lowest current should be at 0 V. We detect the lowest current at -1 V at 250 K likely due to the trap carriers within the p-i-n structure which will induce extra current and requires a higher voltage to balance the extra current and obtain the total lowest current [7]. We have added more detailed description on this in the revised manuscript.

In page 11, the authors mentioned that "As the transmission spectrum of the grating coupler is quite narrow, all the dynamic detection measurements are performed at a wavelength of 1320 nm". Transmission characteristics grating coupler should be added. Moreover, broadband spectral response is an important factor for a photodetector, can the authors provide spectral response of this photodetector? Additionally, there is just one grating coupler in a device, how does the authors define optimal coupling efficiency of the grating coupler? The accurate coupling efficiency is important for responsivity.

Thank you very much for the reviewer's suggestions! The transmission characteristics of the grating coupler are shown in Fig. S6 in the supplementary file. We agree with the reviewer that a broadband spectral response is an important factor for a photodetector. Due to the limit of the grating couplers with different wavelengths, we characterized the free-space spectral response of the photodetector. The spectral dependent IV and spectral dependent current are shown in Fig. S7 in the supplementary material. We compared the spectral dependent current with the absorption coefficient of 53% InGaAs taken from literature. The spectral dependence of the p-i-n device complies well with the expected response InGaAs 53% between 1200 nm and 1600 nm.

We have waveguides on the same chip with various lengths from 300 μm to 10 mm, which have grating couplers on both sides. We measured the waveguides with different lengths and got a transmission loss of 5 dB/mm (results shown in Fig. S6a). We also have 300 μm waveguides with grating couplers on both sides next to each group of active devices. They show uniform transmission loss of 12 dB at 1320 nm (results shown in Fig. S6b). This translates to a 6 dB per coupler in the devices we demonstrated, and we use this value to calibrate the responsivity data shown in Fig. 5 in the paper.

We obtained the optimal coupling efficiency on the waveguide with couplers on both sides (one can be seen from the microscope image shown in Fig. 1 a-6). We inject light from one side and detect from the other side and plot the transmission curves in Fig. S6 b, then determined the optimal coupling efficiency and use this value (6dB) for calculating the responsivity.

A more direct reference to the supplementary material where this information can be found has been added.

Whereas the coupling from the grating coupler to the waveguides as well as the transmission loss of the waveguides can be extracted in this way, we do not have a means for extracting the coupling loss from the waveguide to the III-V region. We expect the coupling efficiency here across the gap to be fairly poor and to be the reason for the relatively low responsivity which we observe and would like to optimize this in the future.

In page 11, the authors mentioned that "As we increase the reverse bias, the responsivity of all devices increases with the reverse voltage in two slopes which can be described by the different carrier transport mechanisms". The detailed information of carrier transport mechanisms should be added.

Many thanks for the suggestion. We have revised the manuscript by adding the two carrier transport mechanisms: carrier trapping at the heterojunction interfaces and carrier drifting in the intrinsic region of the junction. These two carrier transport mechanisms result in the two slopes in the responsivity. We added an energy band diagram in Fig. 5(a) with illustration of the two carrier transport mechanisms.

In page 12, the authors mentioned that "T1, T3 and S4 show responsivities of 0.18, 0.08 and 0.21 A/W at -2 V ". Does the device width lead to this performance difference? Device width has an impact on absorption section and RC constant. Is the optimal width of photodetector equal to the width of waveguide? The authors should explain how the device width leads to different optoelectronic performance?

Thank you for the questions. In line with questions from reviewer 2 we have now revised this figure as well as the accompanying text, so that it hopefully better explains the trade-offs. We have added two schematics illustrating the different optical paths in respectively the T-shape and Straight devices. In a Straight device the optical mode travels along the pin-structure. For the dimensions we are looking at here, the mode should be well confined in the Si waveguide, hence a change of width of the detector/waveguide should not significantly influence the measured absolute current, but it would change the current density as this would be calculated over a larger cross-section area. In the Straight case we would expect an increased absorption by the metal contact on top if instead we were to vary the length of the i-region, but as this is determined by the duration of epitaxial growth it cannot be varied on a device-to-device level.

In the T-shape structures the propagating light impinges on the i-region orthogonally from the Si waveguide. In this case the detector width is independent of the waveguide width. If we make the detector wider, we may expect that more light will be absorbed, so that we should see an increase of absolute current, and possibly also in current density depending on the magnitude of such an increase. In reality, however, this device is most likely limited by the coupling from the Si waveguide to the III-V region over the oxide gap.

These considerations were the reason for implementing those two different designs. As we are dealing with nanostructures, we observe in practice that we are most likely limited by scattering, surface recombination and other losses, which may also be width-dependent. Therefore, it does not allow us to distinguish the minor variations which might be caused by different designs. We decided, however, that it would be most correct here to include the full data set and not just that stemming from a single design.

If we compare in more detail the T1, T3 and S4 with device widths of 200 nm, 350 nm and 500 nm show a responsivity of 0.18 A/W, 0.08 A/W and 0.21 A/W respectively. We plotted the responsivity curves of the three devices in Fig. 5(a) to show the typical two slopes in the responsivity response, which we observe on both the device architectures (Straight and T-shape) regardless of the device width.

We added additional discussion in the revised manuscript to clarify these points.

In Fig. 5a, responsivity of T1 and T3 keeps a constant at reverse voltage from -2 to -0.5 V, but responsivity of S4 decreases with reverse voltage from -2 to -0.5 V. The reverse voltage influences the length of depletion layer and the efficiency of carrier extraction. Why the tendency of these three devices is different?

In Fig. 5a, we show responsivities of three devices including both T-shape (T1 device width=200 nm; T2 device width=350 nm) and Straight (S4 device width=500 nm). We can see the responsivity increase with reverse bias in two slopes for the three devices (we see two slopes in all of the devices we characterized). The responsivity of T1 and T3 slightly decrease with reverse voltage from -2 to -0.5 V with a small slope value, while the responsivity of S4 decreases with a much larger slope value in the same reverse voltage range.

We agree with the reviewer that the reverse voltage influences the length of depletion layer and the efficiency of carrier extraction. Based on this effect and also considering the variation of the device structures (formed during the MOCVD growth, that lead to the different material length and facets in n-InP/i-InGaAs/p-InP/p-InGaAs heterojunctions) with different band offset and depletion layer

length. This will not only lead to different slopes for the responsivity but also different turning points for the two slopes. We have added corresponding information in the revised manuscript.

Many thanks for the reviewer's suggestion which are helpful in improving our manuscript.

In Fig 5a, T-shape structure and S-shape structure have different ways of combining the waveguide and the detector (resulting in different optical power reaching the detector), and the width of absorption region of T1, T3 and S4 is also different (leading to different absorption rates, transport efficiency, and transport time). As we can see, the only variable between T1 and T3 is the device width. Please explain the phenomenon: why the larger the device width, the lower the responsivity? On the other hand, there are too many variables between S4 and other two devices to form an effective comparison. But it can be seen that the relationship between responsivity and bias voltage of S4 is quite different from that of T1 and T3. Please explain the influence of the two patterns on the above relationship.

We compared responsivity data for different architecture and device width which are not shown here, and we cannot make the conclusion that the larger device width, the lower the responsivity. We also see from Fig. 5d that even for the same device width (comparing T1 and T2), the responsivity has a large variation, and as mentioned in the reply to earlier comments the relationship between device geometry and performance is very complex.

We prefer to keep these data in this figure to show that the different architectures with various device width all work as detectors although we observe a significant variation in dark current, responsivity and frequency. We cannot with confidence draw further conclusion from these data. Further research on optimizing the growth, waveguide coupling, and device design would be needed, which is beyond the scope of this work showing the novelty of different approaches for direct growth of waveguide-coupled nanowire photodetectors. We have now revised Fig. 5 as well as the accompanying text in order to better explain the trade-offs of the two device structures.

In page 12, the authors gave the simulation results and mentioned that "the amount of light absorbed in the i-region lies between 10 % and 20 % which is in good agreement with the measured value". The author should give the simulation results of T-shape and S-shape respectively and explain their differences. In addition, the authors should summarize which structure is better, T-shape or S-shape, and tell readers the answer.

Many thanks for the reviewer's suggestions. We plotted the simulation results of the T-shape and straight (S-shape, we prefer to use straight other than S-shape in the paper to avoid misleading the readers that the shape is like the letter "S",) devices in Fig. 9S in the supplementary. We also added the simulation data for discussion and comparison of the T-shape and straight devices in the text of the revised manuscript. Although a higher absorption is seen from the i-region in the T-shaped device, it shows much lower metal loss and slightly lower p-InGaAs absorption compared to the straight device. Considering all the different sources of absorption from the i-InGaAs, metal and p-InGaAs regions, the simulation results suggest that the T-shaped device would have lower loss. We have added a corresponding recommendation in the manuscript.

In Figure 5d-e, the authors showed the responsivity and bandwidth of T-shape and S-shape as a function of device width. An obvious rule is ignored. The responsivity of S-shape structure increases with the increase of the device width, but the bandwidth decreases. The above phenomenon can be explained: the increase in the width of the absorption region brings about a

larger light absorption area (larger responsivity), but it prolongs the carrier transport time (smaller bandwidth). However, the above dependencies of T-shape structure are opposite to that of S-shape structure. Please explain the reason for the different relationship between the responsivity/bandwidth and device width of T-shape and S-shape structures.

We agree with the reviewer that the device width increase will result in a larger light absorption area which is reasonably expected for a larger responsivity, however, for the T-shape devices a larger width corresponds to a longer photon path, and a larger area also results in a higher capacitance, both of which might result in smaller bandwidth. This is a good explanation for the opposite behavior of the responsivity with bandwidth if we compare the devices with same device width regardless of the device architecture. Many thanks for the reviewer's suggestions! We add more information to Fig. 5 and we have modified our manuscript accordingly with more in depth explanation.

In page 12, the authors mentioned that "those devices showing a higher responsivity tend to register a lower f3dB". However, S3 doesn't follow this rule. S3 has a high responsivity and high bandwidth at the same time. The authors need to explain this.

If we compare the responsivity and bandwidth within the same architecture with same device width, we will see an opposite trend. (Meaning if we compare the responsivity of S3 with S4, S4 with higher responsivity registers a lower bandwidth comparing the bandwidth of S3). We have modified our description to make this more clear.

Reviewer #2 (Remarks to the Author):

An interesting approach has been shown by this paper with very good results on III-V photodiodes monolithically integrated on Si. One of the major problem for III-V growth on silicon is defect generation. In this paper, the author did not discuss in detail about this. Can the authors add one paragraph to discuss this issue please? Although good individual device has been demonstrates, how is the yield? Recently, very good quantum dot lasers have been demonstrated on silicon, can the authors discuss the possible route to add lasers on this platform please.

Thank you very much for the reviewer's comments and suggestions. The defect generation in the III-V growth on silicon is an important issue that not only influences the material quality but also affects the device performance both as detector and emitter. We have added paragraphs to discuss the defect issue in the introduction part.

"Due to the lattice mismatch between Si and III-V, defects will arise at the hetero-interface. Traditionally, they can be gradually filtered out by buffer layers and defect-stopping-layers as it is common in direct InP-based epitaxy on Si [8], or they can be mediated by growth from trenches exposing (111) facets [9]. Using these methods excellent devices have been demonstrated, but integration with waveguides remains difficult.

In nanowire growth one relies on a small interface for nucleation between the III-V and Si [10, 11], where defects remain confined near the interface and no propagating dislocations are formed, whereas stacking faults and twins are quite common in nanowire growth.

TASE growth is similar to nanowire growth in that we limit the nucleation site to avoid dislocations, whereas the geometry is determined by the template design rather than the growth conditions. The defects in our TASE grown structures can be localized to the small interface between the Si and III-V seed and result in high quality III-V elsewhere in the template. This is also confirmed by extensive

TEM investigations, where we generally do not observe dislocations in these p-i-n structures. The grown materials are mono-crystalline with an epitaxial relationship to Si. Stacking faults are common, but these should have a minor impact on electrical and optical properties.

The high quality of the material has been demonstrated originally for electronics applications where we could measure mobilities comparable with other III-V films [12-14], and more recently for monolithic optically pumped InP emitters with performance comparable to that of identical devices fabricated by direct wafer bonding [15]. “

Based on those results we are confident that high material quality can be achieved by this method. Nevertheless, with respect to yield, we have not worked to optimize that. Our prime goal in this work is to study different types of devices and geometries in our design, and to demonstrate concepts for high-speed photodetection. We are operating in a shared research facility where we work with different materials and devices. Thus, the cleanliness of our processes is not comparable to that achievable in a commercial process line. Cleanliness is the main factor when assuring homogeneous nucleation, which controls yield and device uniformity.

Lastly, facet control during growth will be essential in achieving identical devices. We do observe variations in the growth of the active material among devices because of the different growth rates of different facets. We as well as other groups are working on gaining increased knowledge in this area specific for TASE growth.

In the long run, we are working on the integration of both detectors and lasers on the same chip. In our group, we have realized InP-on-Si(111) microdisk lasers and lasers based on hybrid III-V/Si photonic crystal structures on SOI [16]. The coupling from the 1D PhC emitters to the waveguide coupled photodetectors demonstrated here, should be straightforward as they are implemented in the same plane and using the same growth conditions. We have added corresponding information in the discussion part.

Reviewer #3 (Remarks to the Author):

The author present a photodiode by using a template assisted selective epitaxy and characterize it as a photo detector and a led. The manuscript provided enough data to show the good quality of the epitaxy, characterize the device as emitter and a photodetector. The performance of the detector is decent. However, the template assisted selective epitaxy method is reported before, so it's not consider a novelty in this manuscript. The device working as a LED doesn't show extra novelty nor significance of this device. In addition, the manuscript lacks of comments/comparisons of the performance of the detector and the LED to literature that made by other integration method. It's a good paper to elsewhere but might be not with high impact enough to be published in Nature Communication.

Thank you for the critical assessment of the manuscript, we hope to have adequately addressed the concerns raised in the following replies. For the novelty of this work, as far as we learned, our manuscript presents the first demonstration of waveguide coupled high-speed III-V in-plane heterostructure photodetectors monolithically integrated on Si. The photodiodes show very good performance as photodetector with a cutoff frequency f_{3dB} exceeding 70 GHz and successfully demonstrate data reception at 50 GBd with OOK and 4PAM. These values are, to the best of our knowledge, the highest reported on III-V nanowire photodetectors on Si. Meanwhile, the diodes can

also perform as a LED with emission centered at 1550 nm. By using the same approach for the integration of the detector and the emitter and an integration technique which enables heterojunctions along the growth direction, this approach can be extended to an all-optical high-speed link on Si without the need for evanescent coupling. We agree with the reviewer, that there probably are standalone devices on bonded platforms which outperform the detectors in terms of data reception and dark currents. However, compared to previous demonstrations of III-V nanowires photodetectors on Si [17-19], the performance is very competitive and exceeds previously demonstrated cut-off frequencies. Unlike previous demonstrations [4], we do not rely on pick-and-place methods, multi-level coupling or regrowth or diffusion for integration of doping profiles. We can leverage the self-alignment with nm precision of passive and active components and the in-situ growth of heterojunctions, thus the novelty is the joint local integration on silicon and the potential to offer significant bandwidth increase in the future and we therefore believe that such a first breakthrough should be communicated in a journal like Nature communications.

In addition, the manuscript lacks of comments/comparisons of the performance of the detector and the LED to literature that made by other integration method.

We have added comments/comparisons of performance of detectors and emitters by different integration methods in the introduction part. And we have added a table to address this in the paper.

Some other questions listed below:

1. It would be good the show clearly how the opening in the oxide is located and dimensioned that for the epitaxy.

Thank you very much for the suggestion. We have added the information on the template opening (position and dimension) in Fig. 1 a-3 in the revised manuscript.

2. In the static characterization of the photodetectors, there are summary charts for dark current, responsivity and frequency. However, I would say that the device design set is not completed. It's a mess for readers.

We plot the summary of the dark current, responsivity and bandwidth of detectors with different architecture and device width to give an overview of how these detectors perform. We agree with the reviewer that the device design has not yet been fully optimized and there is still work remaining in terms of improving both material growth and fabrication process. We have now tried to improve the figure and accompanying text to improve clarity for the reader, in line with the responses to the other two reviewers. Although we cannot always draw clear conclusions, we still believe it is good to show with the different characteristics and the summary chart.

We appreciate the comment and suggestions from the reviewers. Thank you very much for reviewing our manuscript!

Reference

1. Borg, M. et al. Facet-selective group-III incorporation in InGaAs template assisted selective epitaxy. *Nanotechnology* 30, 084004 (2019).
2. Kunert, B. et al. How to control defect formation in monolithic III/V hetero-epitaxy on (100) Si? A critical review on current approaches. *Semicond. Sci. Technol.* 33, 093002 (2018).

3. Qiang, L. & Lau, K. Epitaxial growth of highly mismatched III-V materials on (001) silicon for electronics and optoelectronics. *Prog. Cryst. Growth Charact. Mater.* 63, 105-120 (2017).
4. Baumgartner, Y. et al. High-speed CMOS-compatible III-V on Si membrane photodetectors. *Opt. Express* 29, 509-516 (2021).
5. Shim, J. & Shin, D. Measuring the internal quantum efficiency of light-emitting diodes: towards accurate and reliable room-temperature characterization. *Nanophotonics* 7, 1601-1615 (2018).
6. Olivier, F., Daami, A., Licitra, C. & Templie, F. Shockley-Read-Hall and Auger non-radiative recombination in GaN based LEDs: A size effect study. *Appl. Phys. Lett.* 111, 022104 (2017).
7. Wen, P. et al. Identification of degradation mechanisms of blue InGaN/GaN laser diodes. *J. Phys. D: Appl. Phys.* 48, 415101 (2015).
8. Shi, B. et al. 1.55 μm room-temperature lasing from subwavelength quantum-dot microdisks directly grown on (001) Si. *Appl. Phys. Lett.* 110, 121109 (2017).
9. Wan, Y. et al. 1.3 μm submilliamp threshold quantum dot micro-lasers on Si. *Optica* 4, 940-944 (2017).
10. Tomioka, K., Yoshimura, M. & Fukui, T. A III-V nanowire channel on silicon for high-performance vertical transistors. *Nature* 488, 189-192 (2012).
11. Guo, J. et al. Growth of zinc blende GaAs/AlGaAs heterostructure nanowires on Si substrate by using AlGaAs buffer layers. *J. Cryst. Growth* 30, 359(2012).
12. Schmid, H. et al. Template-assisted selective epitaxy of III-V nanoscale devices for coplanar heterogeneous integration with Si. *Appl. Phys. Lett.* 106, 233101 (2015).
13. Cutaia, D. et al. Complementary III-V heterojunction lateral NW tunnel FET technology on Si. In 2016 IEEE Symposium on VLSI Technology, Honolulu, HI, 2016, pp. 1-2 (2016).
14. Czornomaz, L. et al. Confined epitaxial lateral overgrowth (CELO): A novel concept for scalable integration of CMOS-compatible InGaAs-on-insulator MOSFET on large-area Si substrates. In 2015 Symposium on VLSI Technology, pp. T172-173 (2015).
15. Mauthe, S. et al. InP-on-Si optically pumped microdisk lasers via monolithic growth and wafer bonding. *IEEE J. of Sel. Top. In Quantum Electron.* 25, 8300507 (2019).
16. Mauthe, S. et al. Hybrid III-V silicon photonic crystal cavity emitting at telecom wavelengths. *Nano Lett.* 20, 8768-8772 (2020).
17. Chen, R. et al. Nanophotonic integrated circuits from nanoresonators grown on silicon. *Nat. Commun.* 5, 4325 (2014).
18. Takiguchi, M. et al. Hybrid nanowire photodetector integrated in a silicon photonic crystal. *ACS photonics* 7, 3467-3473 (2020).
19. Xue, Y. et al. High-performance III-V photodetectors on a monolithic InP/SOI platform. *Optica* 8, 1204-1209 (2021).

REVIEWERS' COMMENTS

Reviewer #1 (Remarks to the Author):

My comments have been well addressed. Publish as is.

Reviewer #2 (Remarks to the Author):

The authors addressed most of my concerns in the revised manuscript, although it is not clear on how can these devices for massive production on Si photonics and integrated with other III-V devices. If this point cannot be clearly addressed, it will affect the impact of this paper. Although these results are very good, it could just be incremental.

Reviewer #3 (Remarks to the Author):

The other responded to my and other reviewers' comments well and made a lot of modifies to the manuscript. The author explained the novelty in the response letter very well. It would be great to express a little more of the novelty or the impact in the abstract, introduction, summary or even in the title to distinguish the work from previous. The added benchmark table clearly shows that the device performance is standing out which strengthened the manuscript. I suggested to put the table at the end of the paper probably before discussion part such that allow readers first to learn your work and then get a table to show that where the work is. One more thing is that it would be great to comment the template assist selective epitaxy method in the discussion part regarding to the process feasibility in real manufacturing, any other disadvantages, or what are the advantages than other integration methods like bonding, regrowth on bonded template, etc.

Dear Reviewers,

Thank you very much for your second revision and suggestions! We appreciate your positive comments and constructive suggestions and we revised our manuscript accordingly. All the changes are highlighted in red in the Track mode in the accompanying manuscript. Below are our point-by-point responses to the comments and suggestions raised by the reviewers.

Reviewer #1 (Remarks to the Author):

My comments have been well addressed. Publish as is.

Thank you very much for the reviewer's revision and positive comments!

Reviewer #2 (Remarks to the Author):

The authors addressed most of my concerns in the revised manuscript, although it is not clear on how can these devices for massive production on Si photonics and integrated with other III-V devices. If this point cannot be clearly addressed, it will affect the impact of this paper. Although these results are very good, it could just be incremental.

Thank you very much for the reviewer's comments! In line with suggestions from reviewer 3, we have now added description on the process feasibility in real manufacturing and how to achieve massive production of these devices on Si photonics as well as integrated with other III-V devices. In the past, we demonstrated that we could transfer the TASE technology to a semi-commercial technology platform within IBM. In that case, it was used to demonstrate nanosheet InGaAs transistors on Si for advanced nodes. We have not yet demonstrated a comparable integration level for photonic devices, but conceptually this should certainly be possible. We now reference this work, which was a highlight paper at IEDM in 2018, and have added the following description to the manuscript.

“We previously demonstrated the successful transfer of the TASE concept to an advanced 200 mm process line within IBM, where it was used to demonstrate nanosheet InGaAs FinFETs on Si with a 10 nm channel thickness and state-of-the-art performance [31]. The successful integration of logic devices based on the same technology and for large wafer scale, is promising for also achieving large-scale integration for photonic integrated circuits in the future.”

Reviewer #3 (Remarks to the Author):

The other responded to my and other reviewers' comments well and made a lot of modifies to the manuscript. The author explained the novelty in the response letter very well. It would be great to express a little more of the novelty or the impact in the abstract, introduction, summary or even in the title to distinguish the work from previous. The added benchmark table clearly shows that the device performance is standing out which strengthened the manuscript. I suggested to put the table at the end of the paper probably before discussion part such that allow readers first to learn your work and then get a table to show that where the work is. One more thing is that it would be great to comment the template assist selective epitaxy method in the discussion part regarding to the process feasibility in real manufacturing, any other disadvantages, or what are the advantages than other integration methods like bonding, regrowth on bonded template, etc.

Thank you very much for the reviewer's comments and suggestions! We added the “high-speed” in the abstract as well as the summary to address more impact of this paper.

We moved the table to the end of the results part and before the discussion part as the reviewer suggested. We also added more descriptions regarding the process feasibility in real manufacturing and the advantages compared with other integration methods in the discussion part.

We appreciate the comments and suggestions from the reviewers. Thank you all again for reviewing our manuscript!

In addition, we also made several minor changes to better meet the requirements for the publication on Nature Communications. These changes are highlighted in blue in the Track mode.

They are listed below:

1. We modified the figure legends with more detailed descriptions on each panel and the symbols used.
2. We updated the reference ordering following the guidelines, first throughout the text and then in table.
3. We revised the frequency at -3dB of T3 in Fig. 5e with 70 GHz, which is obtained from the RF response shown in Fig. 6c.
4. We modified the text font in figures to better fit the publication requirements.
5. We modified a bit on the Table and moved the “pick & place” to the device integration column.

All the above changes are minor changes to meet the requirements in the author guidelines and to make better understanding for the readers.